# Separated or joint models of repeated multivariate data to estimate individuals' disease trajectories with application to scleroderma

**Ji Soo Kim** [1,2]*, **Ami A. Shah**[1], **Laura K. Hummers**[1], **Scott L. Zeger**[2]

**1** Division of Rheumatology, Department of Medicine, Johns Hopkins University School of Medicine, Baltimore, Maryland, United States of America, **2** Department of Biostatistics, Johns Hopkins Bloomberg School of Public Health, Baltimore, Maryland, United States of America

* jkim478@jhu.edu

**Data availability statement:** Data cannot be shared publicly because they contain potentially identifying or sensitive patient information. Data are available upon reasonable request from the Johns Hopkins Scleroderma Center Research Registry (contact Center Research Data Manager Adrianne Woods at awoods9@jhmi.edu) and can be shared with appropriate IRB approvals and institutional data sharing agreement.

## Abstract

Estimating a patient's disease trajectory as defined by clinical measures is an essential task in medicine. Given multiple biomarkers, there is a practical choice of whether to estimate the joint distribution of all biomarkers in a single model or to model the univariate marginal distribution of each marker separately ignoring the covariance structure among measures. To fully utilize all trajectory-relevant information in multiple longitudinal markers, a joint model is required, but its complexity and computational burden may only be warranted when joint estimates of trajectories are substantially more efficient than separate estimates. This paper derives general expressions for the inefficiency of univariate or "separated" estimates of population-average trajectories and individual's random effects as compared to the fully efficient multivariate or "combined" estimates. Then, in two settings: (1) a general bivariate case; and (2) our motivating clinical case study with 5 measures, we find that separated estimates of fixed effects are nearly fully efficient. However, joint estimates of random effects can be meaningfully more efficient for measures with substantial missing data when other strongly correlated measures are observed more frequently. This increased efficiency of the joint model derives more from joint shrinkage of random effects in multivariate space than from improved estimates of the subject-specific trajectories obtained when accounting for correlations in measurements. These findings have application to a diverse array of chronic diseases where biomarkers' trajectories guide clinical decisions.

## Introduction

Estimating a patient's trajectory in a space defined by multiple clinical measures is an essential task in medicine. In some problems, a goal is to find a lower-dimensional summary of measures that parsimoniously represents the trajectory. But in the autoimmune disease application that motivates this work, each of the original measures is important to clinical decisions and must be retained.

**Funding:** This work was supported in part by the Jerome L. Greene Foundation, the Johns Hopkins inHealth initiative, the Scleroderma Research Foundation, the Nancy and Joachim Bechtle Precision Medicine Fund for Scleroderma, the Manugian Family Scholar, the Donald B. and Dorothy L. Stabler Foundation, the Chresanthe Staurulakis Memorial Fund, and NIH/NIAMS (P30AR070254, R01AR073208, K24AR080217).

**Competing interests:** The authors have declared that no competing interests exist.

Systemic sclerosis (scleroderma) is an autoimmune disease characterized by dysregulation of the immune system and damage to multiple organ systems, including the skin, heart, lungs, kidneys, gastrointestinal tract, and blood vessels [1]. Although relatively rare, scleroderma is a one of 80 related autoimmune diseases that, in aggregate, comprise the third most prevalent set of chronic diseases after cancer and heart disease [2]. All organs must be monitored to determine appropriate treatment for an individual [3,4]. There is heterogeneity among patients in their clinical manifestations, response to treatment, rate of disease progression, and survival [5]. Therefore, clinicians seek to accurately measure each patient's current disease state and rate of progression or "trajectory" for each organ.

From a statistical perspective, we obtain multivariate longitudinal measures at irregularly-observed times for a cohort of patients. Some measures are easily obtained; others demand considerable resources. Joint and univariate, marker-specific models are widely used to estimate disease trajectories [6,7]. This paper focuses on quantifying the efficiency in estimating fixed and random effects of the joint or "combined (C)" model in comparison to marker-specific or "separated (S)" models. We ask under what circumstances the additional computational and statistical burden of the combined model add substantial value when estimating individual patients' trajectories for all biomarkers. Although the combined models can result in slower convergence and greater challenge in obtaining reliable parameter estimates, it estimates across-measure correlations which is a key scientific question in some applications and cannot be estimated by fitting the separated models. We derive general formulae measuring the relative efficiency for fixed effects, random effects and for predicted values. We quantify the inefficiency of separated models for the individual patient and describe its association with patient's pattern of observations.

The scientific focus on biomarker trajectories implies that the main predictors for both the fixed and random effects are smooth basis functions of time and/or their interactions with baseline patient characteristics. As is well known in other efficiency studies, sharing predictors across regressions has important implications for the efficiency of separated relative to combined models as discussed below.

## Statistical models of trajectory

The linear mixed model (LMM) is widely used to describe changes in a single approximately-Gaussian longitudinal outcome over time. LMMs yield valid inferences about trends by accounting for the autocorrelation among repeated measures of the same subject; they support estimation of subject-specific random effects while naturally handling irregularly spaced or/and unbalanced data [7,8]. Harville (1976) and Harville (1977) [9,10] first applied the Gauss-Markov theorem [11] to the statistical framework for the LMM and showed that its random effects estimators are the best unbiased linear predictors (BLUP) when the covariance parameters are known. The multivariate linear mixed model (MLMM) is an extension of the LMM for the analysis of multiple outcomes [12–15]. Given multivariate longitudinal observations measured for individuals, fitting a separate LMM for each outcome or fitting a single MLMM are both common [16–19]. The separated LMMs approach estimates the population and individual trajectories of each outcome independently of the others, while the MLMM additionally captures the between-measure correlations induced by correlated random effects and random error terms.

## Inefficiency when ignoring correlation

In the time series context, Bloomfield and Watson [20] derived expressions for the maximum inefficiency of ordinary compared to general least squares as a function of the design matrix

and residual variance matrix. A similar idea was explored much earlier by Tukey [21] who quantified the maximum inefficiency caused by using a misweighted mean as compared to the optimally weighted mean.

The gain in efficiency from using a joint model has also been studied for cross-sectional data under the "seemingly unrelated regressions" (SUR) framework [22]. A SUR comprises a set of linear regressions where each equation describes the relationship between a different outcome and its associated predictor variables. Zellner showed that joint coefficient estimation by general least squares (GLS) is asymptotically more efficient compared to separate regressions by ordinary least squares (OLS) and that the efficiency increases as the error terms from different equations become more cross-correlated and as the predictor variables in different equations become less correlated. An important special case is that estimates from separate regressions by OLS are fully efficient when the predictors for each outcome are the same, regardless of the degree of correlation among the outcomes [22]. Oliveira and Teixeira-Pinto further investigated the case in which some predictors are shared across the outcomes while others are outcome-specific and showed that the estimates for the regression parameters of the shared predictors are fully efficient while those of outcome-specific predictors have greater efficiency when a joint model is fit [23].

## Objectives

In this paper, we study the inefficiency of separated LMMs relative to the combined MLMM in the longitudinal data setting. As in previous work on multivariate regression and on time series, we consider the inefficiency in estimating regression coefficients of separated models. However, motivated by our clinical application, an additional focus of this paper is on individual's trajectories as represented by the random effects in the mixed models. We work under the assumption that missing data are missing at random (MAR) [24]. For estimation of the fixed effects parameters, we first consider whether the SUR conditions under which OLS is fully efficient can be satisfied by a MLMM. We then derive expressions for the inefficiency of the outcome-specific LMM estimates relative to the MLMM ones. Focusing on the random slopes (trajectories), we examine the cause and degree of imprecision in two cases: (1) a general two-biomarker problem and (2) our motivating clinical case study of scleroderma trajectories with five biomarkers.

## Efficiencies of separated versus combined models

### Notation

Let $Y_{ijk}$ be the observed value for the $k$th measure for person $i = 1, \dots, m$ at the jth visit $j = 1, \dots, n_{ik}$, at time since onset $t_{ijk}$. Let $Y_{ik}$ be the vector of $Y_{ijk}$ for $j = 1, \dots, n_{ik}$, $X_{ik}$ and $Z_{ik}$ are $(n_{ik} \times p_k)$ and $(n_{ik} \times q_k)$ known matrices of full rank, and $\beta_k$ and $b_{ik}$ are $p_k \times 1$ and $q_k \times 1$ measure-specific vector of parameters for the fixed and random effects. Let $n_i = \sum_{k=1}^{K} n_{ik}$ be the total number of observations for person $i$ and let $e_{ik}$ be the measure-specific, within-subject error term.

With these definitions, the multivariate linear mixed effects model is written as $Y_i = X_i\beta + Z_ib_i + e_i$, $i = 1, \dots, m$ where $\beta = (\beta_1^T, \dots, \beta_K^T)^T$, $Y_i = (Y_{i1}^T, \dots, Y_{iK}^T)^T$, $X_i = \bigoplus_{k=1}^{K} X_{ik}$, $Z_i = \bigoplus_{k=1}^{K} Z_{ik}$, and $\bigoplus$ denotes the Kronecker sum. We assume $b_i = (b_{i1}^T, \dots, b_{iK}^T)^T \overset{ind}{\sim} N_{Kq}(0, D)$, $e_i = (e_{i1}^T, \dots, e_{iK}^T)^T \overset{ind}{\sim} N_{n_i}(0, \Sigma_i)$. Letting $Y = (Y_1^T, \dots, Y_m^T)^T$, $X = (X_1^T, \dots, X_m^T)^T$, $Z = \bigoplus_{i=1}^{m} Z_i$, $b = (b_1^T, \dots b_m^T)^T$, $e = (e_1^T, \dots e_m^T)^T$, $\Gamma = I_m \bigotimes D$ and $\Sigma = \bigoplus_{i=1}^{m} \Sigma_i$, we can write the above model more compactly in the standard linear mixed model form $Y = X\beta + Zb + e$, where $Y \sim G(X\beta, V)$, $V = Z\Gamma Z^T + \Sigma$ and $b \sim G(0, \Gamma)$, $e \sim G(0, \Sigma)$.

## Defining combined (joint) and separated models

In the specification above, $D$ and $\Sigma_i$ are ($Kq \times Kq$) and ($n_i \times n_i$) positive definite matrices, respectively. The K ($q \times q$) and ($n_{ik} \times n_{ik}$) measure-specific block diagonal matrices for $D$ and $\Sigma_i$ represent within-measure covariance of random effects and random errors, respectively. The off-block diagonals of $D$ and $\Sigma_i$ represent the covariances of random effects and random errors across measures. If the off-diagonal submatrices are set equal to zero, then the mixed effects model of K measures reduces to K univariate mixed effects models. We call this the "separated" model in contrast with the model with the unrestricted $D$ and $\Sigma_i$ that is called the "combined" model.

For the separated model, $Y_i = X_i\beta_S + Z_ib_{Si} + e_i$, $i = 1, ..., m$ where $b_{Si} \sim G_{Kq}(0, D_S)$, $e_i \sim G_{n_i}(0, \Sigma_{Si})$ so that $Y \sim G(X\beta_S, V_S)$, $V_S = Z\Gamma_S Z^T + \Sigma_S$, $\Gamma_S = I_m \bigotimes D_S$, and $\Sigma_S = \bigoplus_{i=1}^{m} \Sigma_{Si}$.

For the combined model, $Y_i = X_i\beta_C + Z_ib_{Ci} + e_i$, $i = 1, ..., m$ where $b_{Ci} \sim G_{Kq}(0, D_C)$, $e_i \sim G_{n_i}(0, \Sigma_{Ci})$, $Y \sim G(X\beta_C, V_C)$, $V_C = Z\Gamma_C Z^T + \Sigma_C$, $\Gamma_C = I_m \bigotimes D_C$, and $\Sigma_C = \bigoplus_{i=1}^{m} \Sigma_{Ci}$. To simplify the notation, let $W_S = V_S^{-1}$, $W_C = V_C^{-1}$ and $W_{S_i} = V_{S_i}^{-1}$, $W_{C_i} = V_{C_i}^{-1}$ in following sections.

## Separated models and seemingly unrelated regressions (SUR)

The fixed effects estimates $\hat{\beta}_C$ from the combined model are generalized least squares (GLS) estimates first described in Aitken [25]. They are therefore the best linear unbiased estimator (BLUE) so that the variance of $\hat{\beta}_S$ is greater than or equal to the variance of $\hat{\beta}_C$. There are, however, two situations where the separated models' fixed effects estimates are fully efficient as originally discussed in Zellner [22], summarized in S1 Supporting materials. The first is a trivial case when the cross-measure covariances of error terms are zero, where the combined model is equivalent to the separated models. The other case is when the measure-specific design matrices $X_{ik}$ are the same across all $k = 1, ...K$ measures.

The question is whether, for multivariate linear mixed effects models, the separated models can be fully efficient as occurs on the SUR case? In Supporting Materials S1, we show that the separated models always lose efficiency relative to the multivariate model except when: (1) the cross-measure covariances of error terms and random effects are all zero; (2) $X_{ik}$ are the same for all $k$ and $Z_{ik}$, a sub-matrix of $X_{ik}$, are the same for all $k$.

So, the question remains, how inefficient are the separated models for multivariate cross-sectional responses? Is the inefficiency sufficient to warrant the burden of jointly modeling the outcomes in situations like tracking disease progression where the separated models meet the clinical objectives?

## Comparing estimates of combined and separated models

Our interest lies in quantifying the improvement in efficiency of the combined model relative to the separated model when both can provide valid inferences that address the clinical question in estimating: (1) fixed effects coefficients that represent population average trajectories $\hat{\beta}$; (2) an individual's estimated random effects $\hat{b}_i$ that represent his estimated deviations from the average trajectories; and (3) an individual patient's estimated trajectories $\hat{y}_i$ that are a linear combination of $\hat{\beta}$ and $\hat{b}_i$.

We compute the following ratios of mean squared error (MSE) for each of $\hat{\beta}$, $\hat{b}_i$, and $\hat{y}_i$ from the combined and separated model.

$$\text{MSE Ratio of } \beta = \text{MSE}(\hat{\beta}_C, \beta)/\text{MSE}(\hat{\beta}_S, \beta) \tag{1}$$

$$\text{MSE Ratio of } b_i = E_{b_i}\{\text{MSE}(\hat{b}_{Ci}, b_i)\}/E_{b_i}\{\text{MSE}(\hat{b}_{Si}, b_i)\} \tag{2}$$

$$\text{MSE Ratio of } y_i = E_{b_i}\{MSE(\hat{y}_{Ci}, E(\hat{y}_i|b_i))\}/E_{b_i}\{MSE(\hat{y}_{Si}, E(\hat{y}_i|b_i))\} \tag{3}$$

Formulae for the MSE, variance, and squared bias are presented in Supporting Materials S2.

## Case studies

In two case studies, we examine the inefficiencies resulting from fitting separate LMMs rather than a single MLMM, derived from the general expressions. As detailed below, the first is the general bivariate case with fixed predictors, covariance matrices, degrees of missing data, and simulated missing data patterns in which we can examine the entire space of correlations between the two measures. In this first case, we focus on the inefficiency of the random effects because the fixed effects estimates are close to fully efficient. The second is the motivating scleroderma study in which there are 5 distinct measures where we consider the inefficiency of both the fixed and random effects.

### Bivariate case study

Consider two measures $Y_{i1}$ of length $n_{i1}$ and $Y_{i2}$ of length $n_{i2}$ for subject $i$. Let $b_{i1}$ and $b_{i2}$ be the measure-specific vectors of random effects for $Y_{i1}$ and $Y_{i2}$. With little loss of generality, we simplify the problem by assuming that population regression coefficients $\beta$ are known for the following reasons. First, the separated model is fully efficient in estimating $\beta$ if $Z_i$ is a part of $X_i$ and if $X_{ik}$ is the same across measures and subjects (see S1 Supporting materials). In other cases, we observe that the degree of inefficiency in estimating random effects are numerically similar when estimating $\beta$ as compared to known $\beta$.

When both measures are fully observed, we expect little benefit from fitting the combined model. The real value of fitting the combined model is when one of the variables is poorly determined, either due to missing data or noise in the measurements. We let $Y_{i1}$ be fully observed and $Y_{i2}$ be missing in various degrees to measure efficiency gains for the random effects $b_{i2}$ by fitting the combined model. Our primary interest in this case study is estimating $b_{i2}$, as jointly modeling would only have marginals effects on $b_{i1}$, which is already well determined. We calculate MSE Ratio of $b_{i2}$, MSE ratio that corresponds to the random effects of the second measure, by taking sub-matrices of MSEs in the equation 2. MSEs for the separated and combined models with known $\beta$ are derived in S3 Supporting materials.

Using the formulae in S2 Supporting materials, we investigate the relative contributions to improved efficiency of (1) the degree of missingness, (2) measurement error, (3) heterogeneity in individual trajectories, and (4) across-measure correlations.

**Degree of missingness.** With complete data, the random effects design matrix $Z = Z_{i1} \bigoplus Z_{i2}$ comprises the constant vector for the intercept and a vector of equally spaced times, scaled to range from -1 to 1, for the trajectory. We assume $Z_{i1}$ is fully observed, while a portion $p_{miss}$ of $Z_{i2}$ is randomly missing. When $p_{miss} = 0$, $Z_{i1} = Z_{i2}$, and $n_{i1} = n_{i2}$; when $p_{miss} = 33\%$, $n_{i2} = \frac{2}{3}n_{i1}$. In applications, such a pattern is observed when one measure is more frequently collected than the other. We also consider the case where $Z_{i2}$ is missing in a drop-out pattern, such that we only observe the first $n_{i2}$ observations of the second measure and the rest is lost to follow up.

**Degree of measurement error and heterogeneity in trajectories.** To investigate under which scenario the combined model borrows most strength in estimating $b_{i2}$, we consider three cases based on relative sizes of the variance components.

The random effects covariance matrix $D$ and measurement error covariance matrix $R$ are defined by four correlation variables. Let

$$D = \begin{pmatrix} D_{11} & D_{12} \\ D_{21} & D_{22} \end{pmatrix} = \begin{pmatrix} d_{11} & d_{12} & d_{13} & d_{14} \\ d_{21} & d_{22} & d_{23} & d_{24} \\ d_{31} & d_{32} & d_{33} & d_{34} \\ d_{41} & d_{42} & d_{43} & d_{44} \end{pmatrix}; R = \begin{pmatrix} r_{11} & r_{12} \\ r_{21} & r_{22} \end{pmatrix}, \tag{4}$$

where $D_{kk} = Var(b_{ik})$ and $D_{kk'} = Cov(b_{ik}, b_{ik'})$.

Case A: equal-sized random effect and measurement error variances;
$d_{11} = d_{22} = d_{33} = d_{44} = r_{11} = r_{22} = 1$.

Case B: unequal measurement errors; variance components of the random effects and measurement errors are as in Case A except that $r_{22} = 4$, representing greater measurement error for $Y_2$.

Case C: heteroskedastic random effects and equal measurement error variances; variance components of the random effects and measurement errors are as in Case A except that $d_{44} = 4$. Greater variances are assumed for the random slope variance for $Y_{i2}$ to produce substantial heterskedasticity in $Y_{i2}$ across time.

We assess the degree to which greater measurement error in $Y_{i2}$ (Case B) or increased heterogeneity in trajectories of the second measure (Case C) results in greater efficiency gains for the joint model compared to our reference case of equal-sized random effect and measurement error variances (Case A).

**Evaluation of efficiency at different combinations of within and across-measure correlations.**  For each case, we explore the entire range of across-measure correlation between the random effects $\rho_b$, across-measure correlation of measurement errors $\rho_r$, and within-measure correlations of random intercept and slope for the two measures $\rho_k^1$ and $\rho_k^2$. For simplicity, we report results only for the case $\rho_k^1 = \rho_k^2$. The general formulae in the Supporting Materials S3 can be used when $\rho_k^1 \neq \rho_k^2$.

When exploring the range of correlations and variances summarized above, we ensure the resulting random effects covariance matrix $D$ is positive-semi-definite using a slightly modified version of the spectral decomposition method introduced in Rebonato and Jäckel (2001) [26].

**Gains in efficiencies.**  Fig 1 shows the relative efficiencies for $b_{i2}$ for the combined versus separated models. We see that the greater the absolute values of $\rho_b$ and $\rho_r$, the higher the efficiency gain for the combined model. We observe greater gains when $\rho_b$ and $\rho_r$ are less similar, that is when the two sources of variability introduce correlations of opposing signs. However, in practice, large values with opposite signs for $\rho_b$ and $\rho_r$ are highly unlikely. Hence, the efficiencies are close to one in most practical situations.

In the complete $Y_2$ case (first row of Fig 1), the separated model is nearly full efficient when $\rho_b$ is similar to $\rho_r$ regardless of their magnitude, a result reminiscent of the SUR case. With no missing data, $Z_1 = Z_2$, the situation in the SUR model where OLS is fully efficient. The inefficiency of the separated model increases as $p_{miss}$, the fraction of missing data for $Y_2$, increases. In fact, $p_{miss}$ affects the efficiency more than the correlation parameters over their realistic ranges. This pattern is consistent across different combinations of $n_{i1}, \rho_k^1, \rho_k^2$.

The effects of varying $\rho_b, \rho_r, \rho_k$ on efficiency gain by $n_{i1}, p_{miss}$, and case are presented in Fig 2. The result illustrates that fitting the combined model is particularly advantageous when missingness in $Y_{i2}$ is large. For those with complete $Y_{i2}$ data, the average gains across individuals are minimal, especially for case A. On the individual level, however, we observe combinations of $\rho_b$ and $\rho_r$ with decreased MSE ratios. Assuming greater variance for random measurement error for $Y_2$, 25th percentiles of ratios have greater than 15% decrease in MSE by

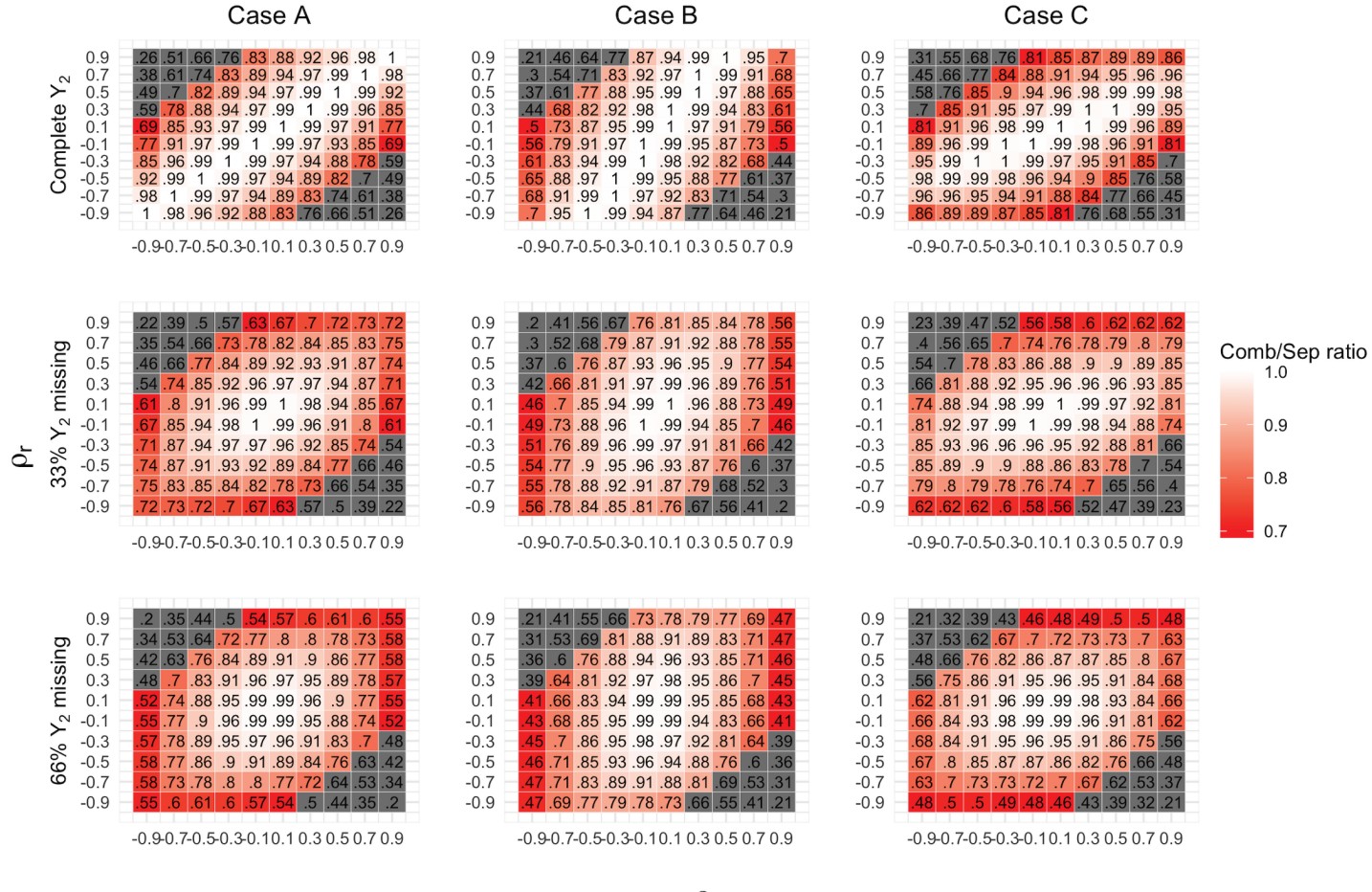

**Fig 1. MSE Ratio of $b_{i2}$ by varying $p_{miss}$, $\rho_b$ and $\rho_r$ under scenarios A, B, and C when $n_{i1} = 6$, $\rho_k = 0$ for randomly missing case.** Cells representing unlikely combinations of $\rho_b$ and $\rho_r$ are colored in grey.

fitting the combined model even with 0% missingness. Additionally, in a special case of the model specification in which we assume random effects are the only source of across-measure correlations, we can still expect greater efficiency gains when $|\rho_b|$ is high.

The gains are much larger when we increase $p_{miss}$ to 33% and 66%. There is heterogeneity in MSE ratios ranging from 0.3 to 1. The variability is more closely related to values of $\rho_b$ and $\rho_r$ than $n_{i1}$, the absolute number of observations. When between-measure correlation is weak, the separated model can be fully efficient even with large $p_{miss}$. The gains are slightly greater in the drop-out missing pattern, but the findings are qualitatively similar (see S4 Supporting materials, S1 Fig, and S2 Fig).

From these results, we conclude that the available information in the measure itself and other correlated measures together determines the benefit from fitting the combined model to estimate individual trajectories. Individuals with rich $Y_{i2}$ data can obtain reasonable trajectory estimates by only modeling $Y_{i2}$, while individuals with sparse $Y_{i2}$ data can achieve substantial reduction in MSE by fitting the combined model if the two measures are highly correlated.

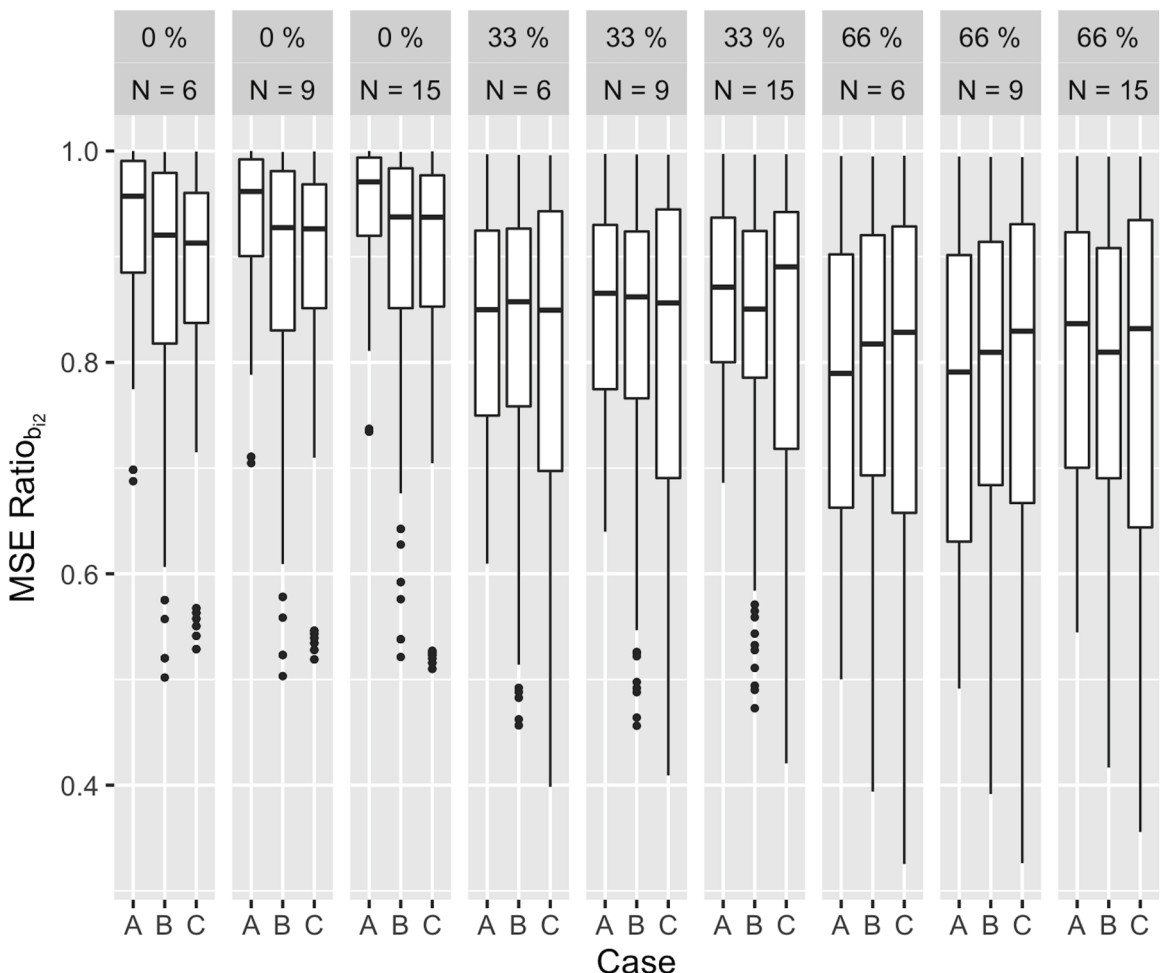

**Fig 2. MSE ratio of $b_{i2}$ evaluated at $p_{miss}$ = 0%, 33%, 66%, $n_{i1}$ = N = 6, 9, 15, $\rho_r$ = −0.9, −0.7, ..., 0.7, 0.9, $\rho_b$ = −0.9, −0.7, ..., 0.7, 0.9, $\rho_k$ = −0.5, 0, 0.5 under cases A, B, and C for randomly missing case.** Only the likely combinations of $\rho_b and \rho_r$ illustrated in Fig 1 are used to create boxplots. Different combinations of $\rho_k^1 and \rho_k^2$ yield negligible difference in outcome, so we only present the case of common $\rho_k$ that takes three values.

## Scleroderma case study

**Sclerodema data.** The Johns Hopkins Scleroderma Center Cohort comprises over 4,000 patients, providing a unique opportunity to study trajectory-focused prediction tools relevant to any chronic disease that manifests in many biomarkers. In scleroderma, clinicians track: pulmonary function measured by the standardized percent predicted forced vital capacity (pFVC) and standardized percent predicted diffusing capacity for carbon monoxide (pDLCO); cardiac function measured by left ventricular ejection fraction (EF) and right ventricular systolic pressure (RVSP); and skin measured by the modified Rodnan skin score (mRSS). Disease onset is defined by the earlier of the onset of Raynaud's phenomenon, reduced blood flow to the fingers, and first non-Raynaud's symptom. In this paper, we study 581 patients who have at least 4 repeated observations for each of the 5 measurements within 40 years since disease onset.

Some measures are collected more frequently than other measures. On average, we see greater numbers of skin and pulmonary measures compared to those of cardiac measures.

This is because pFVC and pDLCO from pulmonary function tests and mRSS from routine clinical visits are more easily collected compared to EF and RVSP from echocardiograms. The summary statistics of the number of observations by measure are shown in Table 1.

All five measures were quantile normalized by mapping their empirical marginal distributions to the Gaussian distribution. Let $Y_k$ be a vector of the observed values from measure $k = 1, ..., 5$. The quantile-normalized vector is obtained by $\Phi^{-1} \circ \hat{G}_k(Y_k)$, where $\hat{G}_k Y_k$ and $\Phi^{-1}$ is the inverse of the standard Gaussian distribution. RVSP and mRSS are transformed by multiplying them by -1 so that an increase in all five measures indicates improved disease status.

**Estimating the separated and combined models.** Because the clinical focus is on patient trajectories, the fixed effects of our model included natural splines of time with 3 degrees of freedom, age of onset, race, sex, skin type, presence of three common autoantibodies, and the interactions of each of the baseline covariates listed above with the natural spline of time. Patient specific intercept and linear time are included as random effects. Standard linear mixed model software including R packages **lme4** [27] and **nlme** [28] can easily fit the separated models. However, in this case-study, the algorithms failed to converge when fitting the combined model despite substantial efforts to tailor the starting values and convergence tuning constants. The combined model with saturated random effects and residual covariances requires estimation of 40 + 10 additional parameters in the random effects and residual covariance matrices, respectively, compared to those of the separated model.

We therefore fit the combined model using the R package **MCMCglmm**[29]. For the fixed effects of both models, we used a diffuse independent Gaussian prior centered around zero with a large variance ($10^8$). Weakly informative inverse-Wishart priors are placed on random effects and residual covariance matrices. Specifically, we set the prior distribution of the random intercepts to have a mode of one and those of random slopes to have the mode of 0.005, with 10 degrees of freedom. The prior distribution of the residual covariance matrix also had mode of one for each measure with 5 degrees of freedom. The degrees of freedom are chosen to make the distributions as diffuse as possible while guaranteeing they are valid inverse-Wishart distributions.

An alternative to fitting the combined model is to estimate the cross-measure covariance parameters using within-measure variance estimates obtained by fitting separated models. Jackson et al. [30] and Chen et al. [31] extended DerSimonian and Laird's univariate method of moments estimator [32] to a multivariate setting to estimate the cross-measure covariance matrix in the random effect model from the measure-specific models. Using such methods and our formulae, we can compute the inefficiency of fitting the separated models without directly fitting the combined model and evaluate whether the combined model should be fit.

**Scleroderma covariances within and across-biomarker.** We compare the assumed covariance structures of the combined and separated models with the estimated covariance matrices. In Fig 3, we show that the combined model captures the within and across-measure

**Table 1. Summary statistics of 581 patients' number of observations**

|                      | pFVC  | pDLCO | EF    | RVSP  | mRSS  |
|----------------------|-------|-------|-------|-------|-------|
| Mean                 | 12.83 | 12.40 | 9.13  | 7.47  | 19.09 |
| Standard deviation   | 6.20  | 6.01  | 3.71  | 3.28  | 7.61  |
| Total number of Obs  | 6136  | 5789  | 4281  | 3281  | 9055  |

correlation patterns quite well; the separated model only captures within-measure correlations. We display the correlations of yearly average values for the first 10 years of follow-up within and among the five measures.

We observe high positive correlation for the two lung measures which suggests that there could be gains in efficiency when modeling the measures jointly. We observe positive correlation between RVSP, mRSS, and the two lung measures; the EF observations appear to be uncorrelated with any other measure including RVSP, the other cardiac measure.

**Comparing bias and efficiency.** Using the formulas derived in Supporting Materials S2, we compare MSE, bias and variance of: (1) the fixed effects estimates $\hat{\beta}_C$ and $\hat{\beta}_S$; (2) random effects estimates $\hat{b}_{Ci}$ and $\hat{b}_{Si}$; and (3) the predicted values $\hat{y}_{Ci}$ and $\hat{y}_{Si}$. All three estimands of interest are functions of the design matrices ($X$ and $Z$) and covariance matrices ($D_C$, $D_S$, $\Sigma_{Si}$, and $\Sigma_{C_i}$). We construct design matrices for each individual using observed times at which the five measurements are taken based on the model described above. From the model, we also estimate the population covariance of the random effects $D_C$ and population residual covariance $\Sigma_{Ci}$. In this section, we use the finite sample posterior estimates of the variances obtained by taking the posterior mean of the MCMC estimates of $D_C$ and $\Sigma_{Ci}$. The variances $D_S$ and $\Sigma_{Si}$ for the separated model are constructed by setting the off– diagonal terms of $D_C$ and $\Sigma_{Ci}$ to be zero.

**Population average trajectory estimation.** In Table 2, we present overall and measure-wise MSE Ratio of $\beta$ obtained from Equation 1. Assuming known variance parameters, the overall MSE in estimating fixed effects is reduced by only 3% when using the combined model compared to fitting the separated model. Since both fixed effect estimates for the separated and combined models ($\hat{\beta}_S$ and $\hat{\beta}_C$) are unbiased (see S2 Supporting materials), the reduction in MSE solely comes from variance reduction.

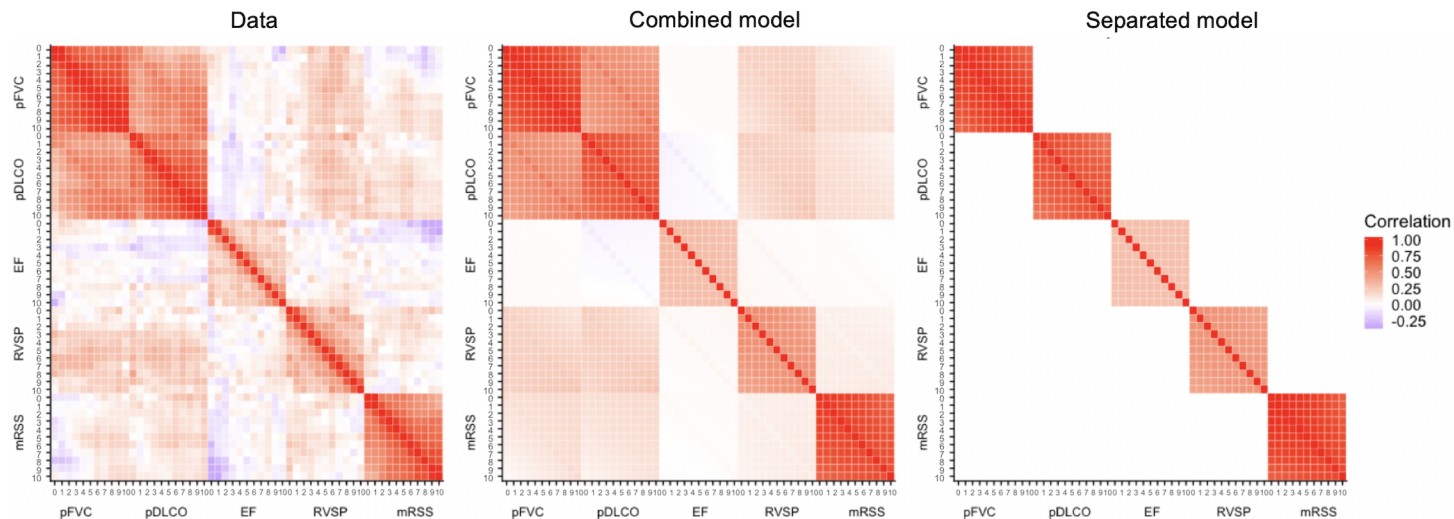

**Fig 3. Pairwise correlations of observations from all patients for 11 years (years 0,...,10 since the disease onset) are calculated and plotted (left) using range of colors from red, white, and blue each representing correlation of 1, 0, and −1, respectively.** The 11 by 11 block matrices on the diagonals shows the degree of correlation in patients' repeated observations over time for each of the five measures. Looking along the block-diagonal, one observes that the two lung measurements and mRSS are highly correlated with their respective past observations, while observations of the two heart measures have less serial correlation. The empirical correlation matrices of the combined and separated models (middle and right) are plotted using the covariance estimates from the two models. The combined model allows correlation among the five measures, while the separated model does not.

**Table 2. Ratio of MSE of overall and measure-wise fixed effects of the combined model to the separated model**

|  | Overall | pFVC | pDLCO | EF | RVSP | mRSS |
|---|---|---|---|---|---|---|
| MSE ratio of $\beta$ | 0.97 | 0.98 | 0.97 | 0.99 | 0.95 | 0.99 |

**Estimating random effects and individual patients' prediction.** Patients' deviations in the level and trend from the average population trajectory is captured by the random intercept and slope estimates. Table 3 presents subject-average MSE ratios from Equation 2 for random effects (MSE Ratio of $b_i$), random intercepts only (MSE Ratio of $b_i^{intercept}$), and random slopes only (MSE Ratio of $b_i^{slope}$). Equation 3 is a similar expression for predicted values (MSE Ratio of $y_i$). Estimating random effects and predicted values from the combined model is most advantageous for RVSP. Overall, the mean gains in MSEs are minimal.

**Heterogeneity in bias and efficiency gains by patient.** Random effects estimates are a linear combination of patient-specific level and trajectory estimates and their population analogues. Hence, depending on the amount and characteristics of individuals' data, we can expect variation among patients in the MSEs. In Fig 4, we show the measure-specific MSE for estimators of individual patient's fixed effects, random effects, random slope only, and predicted values. The MSEs are transformed onto the log scale; a positive value indicates that the separated model has smaller errors and a negative value indicates that the combined model does.

In panel (a), the five points marking the measure-specific log ratios of MSE are equivalent to the MSE Ratio of $\beta$ in Table 2 transformed to the log scale. The box plots in panels (b)–(d) show the patient-level log ratios for the random effects, random slope estimates, and predicted values for the 581 patients.

The most notable result is that there is sizable heterogeneity for the patient-specific log ratios, especially for pDLCO and RVSP. For RVSP, most patients benefit from fitting the combined model. The gains in pDLCO are substantial for only 25 percent of the patients. The stretched out left tails of the pDLCO MSEs indicate that a small fraction of patients are estimated to have over 20% efficiency gains.

**Ethics approval and consent to participate.** Data analyzed in this study were obtained from consenting participants in the Johns Hopkins Scleroderma Center Research Registry accessed on February 10th, 2020. This study was approved by the Johns Hopkins Medicine Institutional Review Board (IRB00251593 and IRB00226995). Participants provided written informed consents, and the authors have permission to identify patients during and after data collection for additional data collection. All methods were performed in accordance with the relevant guidelines and regulations.

**Table 3. Average MSE ratios of random effects, random intercept only, random slope only, and predicted values of the combined model to the separated model**

|  | pFVC | pDLCO | EF | RVSP | mRSS |
|---|---|---|---|---|---|
| MSE Ratio of $b_i$ | 0.98 | 0.96 | 0.98 | 0.95 | 0.99 |
| MSE Ratio of $b_i^{intercept}$ | 0.98 | 0.96 | 0.98 | 0.95 | 0.99 |
| MSE Ratio of $b_i^{slope}$ | 0.97 | 0.95 | 0.98 | 0.91 | 0.98 |
| MSE Ratio of $y_i$ | 0.99 | 0.97 | 0.99 | 0.97 | 0.99 |

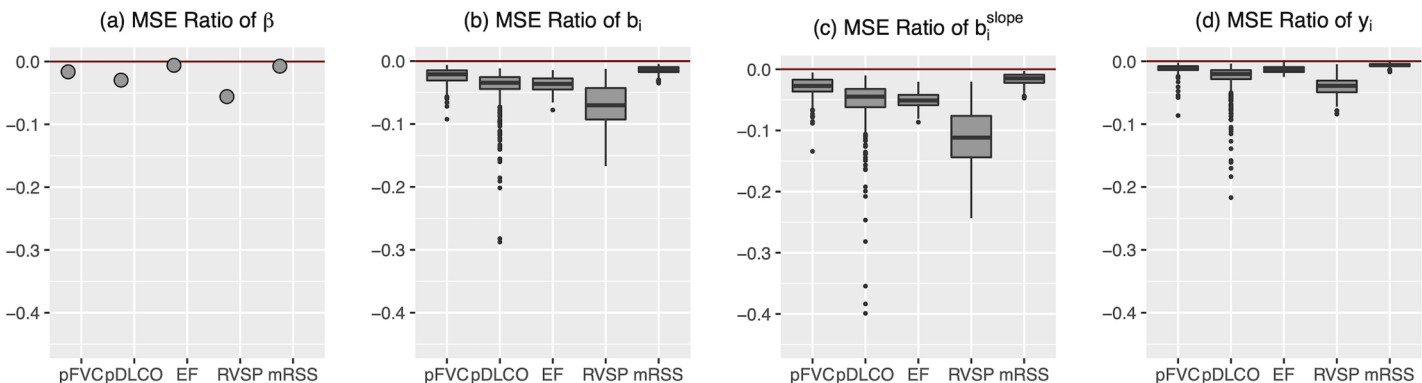

**Fig 4. Log ratio of MSEs of the combined model to the separated model for estimators of individual patient's fixed effects, random effects, random slope only, and predicted values.**

## Discussion

In our application, a patient's disease state is reflected in multiple irregularly spaced longitudinal measures. By selecting and estimating multivariate Bayesian hierarchical models, we estimated smooth individual and population trajectories for each measure/organ systems using noisy and, for many individuals, sparse data. This analysis can further clinicians' understanding of the disease by representing disease progression in multiple dimensions for clinically-defined subpopulations and by quantifying the correlations across measures and time.

We addressed the statistical question of whether, in a simple bivariate case and in our motivating scleroderma example with five biomarkers, fitting a more complex multivariate hierarchical model ("combined model") produces substantially more efficient estimates compared to fitting a set of "separated models," one for each measure. In regression analysis, this question was raised by Zellner [22]. He showed that the coefficient estimation using the GLS [25] is asymptotically more efficient compared to the OLS, and that the efficiency increases as the error terms from different equations become more cross-correlated and as the predictor variables in different equations become less correlated. The OLS estimates are fully efficient when either there is no cross-measure correlation or when the predictors are the same in the regression model for each measure. We show that multivariate linear mixed models are not separable into individual equation systems without efficiency loss except for restrictive special cases. We derived equations to quantify this loss of efficiency for fixed effects, and in one general and one specific case study showed the inefficiency is negligible.

With mixed effect models, there is an additional question of how inefficient are estimates of random effects using separated as compared to combined models. We derived a set of generalized formulae to compare the relative efficiency of individual-level estimates from the fully efficient combined model and the simpler separated models. We defined the MSE for the estimated random effects as the average conditional MSE over the distribution of $b_i$. The conditional MSE is defined as the conditional expected squared difference between the predicted values above and the true value of the random effect.

There is little advantage of the combined model for estimating the fixed effects, when multiple outcome measures are observed at similar or common times. Such patterns are frequently observed in longitudinal studies, including our own case study. The five measures are captured at the same or similar times, making the fixed and random effects design matrices

similar, resulting in separated models being almost fully efficient. For the random effects estimates, however, the degree of advantage of the combined model can be substantial for some individuals. There are sizable gains for those individuals for whom the relative number of observations in the measure of interest is smaller than those in other correlated measures. The degree of efficiency gain increases with the degree of cross-measure correlations.

The increased efficiency of the combined model estimates of individual trajectories derive mainly from multidimensional shrinkage toward the population mean trajectory instead of shrinking in each dimension separately as occurs in separated models. The population average trajectories for the five measures are estimated with near full-efficiency with the separated model. However, the combined model is advantageous as it shrinks the subject-specific measurements towards the population average trajectories in a multivariate space, whereas the separated model shrink within each univariate space. The rate of shrinkage depends on cross-measure correlations and missingness, producing different efficiency loss across subjects.

In terms of bias and variance, the efficiency gain, or the reduction in MSE for the random effects mostly results from reduced bias. For individuals who have only a few data points for a given measure, the data for the measure alone cannot accurately reflect the underlying disease state of the individual. Hence, fitting the separated models results in greater shrinkage towards the measure-specific mean and results in larger bias. The bias is reduced when fitting the combined model, where the random effects estimator borrows strength from data-rich measures.

This framework for comparing the performances of the combined and separated models for the population and individual level estimates can be applied to any setting where the individuals' and population trajectories in higher dimension space need to be estimated. However, it should be noted that the results are drawn assuming Gaussian responses after transformation and missingness at random. The effects of non-Gaussian and non-ignorable missingness on the results are topics for further studies. The results in this paper describe the efficiency costs of misspecifying the covariance structure among the random effects and/or residual errors. Another form of misspecification is by omitting key predictors. In our particular application, this might involve assuming a smooth trajectory for a biomarker when the changes are more acute or immediate. The efficiency results presented in this paper assumed that linear predictors are correctly specified. When the models are misspecified, there is no a priori reason to believe that the effects of misspecification would be more or less for the combined versus separated models.

## Software

Software in the form of R code, together with a sample input data set and complete documentation is available on request from the corresponding author.

## Acknowledgments

The authors thank Professor Antony Rosen, director of the Johns Hopkins inHealth Precision Medicine program, Fred Wigley director of the Johns Hopkins Scleroderma Center, and Aalok Shah for supporting our use of the JH Precision Medicine Analytics Platform.

## Supporting information

**Supporting materials** includes four sections: **S1** Efficiency of fixed effect estimates and seemingly unrelated regressions **S2** Mean squared error and bias-variance decomposition **S3** Mean

squared error and bias-variance decomposition of random effect estimates with known population parameters **S4** Efficiency gains for the random effects in the case of drop-out missing pattern.
(PDF)

**S1 Fig.** MSE Ratio of $b_{i2}$ *by varying* $p_{miss}$, $\rho_b$ *and* $\rho_r$ under scenarios A, B, and C when $n_{i1} = 6$, $\rho_k = 0$ for drop-out missing case. Cells representing unlikely combinations of $\rho_b$ *and* $\rho_r$ are colored in grey.
(TIF)

**S2 Fig.** MSE Ratio of $b_{i2}$ *evaluated at* $p_{miss}$ = 0%, 33%, 66%, $n_{i1} = N = 6, 9, 15$, $\rho_r = -0.9, -0.7, ..., 0.7, 0.9$, $\rho_b = -0.9, -0.7, ..., 0.7, 0.9$, $\rho_k = -0.5, 0, 0.5$ under cases A, B, and C for drop-out missing pattern. Only the "likely combinations" of $\rho_b$ *and* $\rho_r$ illustrated are used to create boxplots. Different combinations of $\rho_k^1$ *and* $\rho_k^2$ yield negligible difference in outcome, so we only present the case of common $\rho_k$ that takes three values.
(TIF)

## Author contributions

**Conceptualization:** Ji Soo Kim, Ami A. Shah, Scott L. Zeger.

**Data curation:** Ami A. Shah, Laura K. Hummers.

**Formal analysis:** Ji Soo Kim, Scott L. Zeger.

**Funding acquisition:** Ji Soo Kim, Ami A. Shah, Laura K. Hummers, Scott L. Zeger.

**Investigation:** Scott L. Zeger.

**Methodology:** Ji Soo Kim, Scott L. Zeger.

**Project administration:** Ami A. Shah.

**Resources:** Laura K. Hummers.

**Software:** Ji Soo Kim.

**Supervision:** Scott L. Zeger.

**Visualization:** Ji Soo Kim.

**Writing – original draft:** Ji Soo Kim.

**Writing – review & editing:** Ji Soo Kim, Ami A. Shah, Scott L. Zeger.

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
