## [Decision Letter · Decision Letter 0]

20 Oct 2024

PONE-D-24-36884Separated or joint models of repeated multivariate data to estimate individuals' disease trajectories with application to sclerodermaPLOS ONE

Dear Dr. Kim,

Thank you for submitting your manuscript to PLOS ONE. After careful consideration, we feel that it has merit but does not fully meet PLOS ONE’s publication criteria as it currently stands. Therefore, we invite you to submit a revised version of the manuscript that addresses the points raised during the review process.

We look forward to receiving your revised manuscript.

Kind regards,

Pinaki Sarder

Academic Editor

PLOS ONE

Journal Requirements:

2. Thank you for stating the following financial disclosure: This work was supported in part by the Jerome L. Greene Foundation, the Johns Hopkins inHealth initiative, the Scleroderma Research Foundation, the Nancy and Joachim Bechtle Precision Medicine Fund for Scleroderma, the Manugian Family Scholar, the Donald B. and Dorothy L. Stabler Foundation, the Chresanthe Staurulakis Memorial Fund, and NIH/NIAMS (P30AR070254, R01AR073208, K24AR080217). 

3. In the online submission form, you indicated that data are available upon reasonable request from the Johns Hopkins Scleroderma Center Research Registry PI, Dr. Ami Shah (Ami.Shah@jhmi.edu).

Reviewers' comments:

Reviewer's Responses to Questions

**Comments to the Author**

1. Is the manuscript technically sound, and do the data support the conclusions?

Reviewer #1: Yes

Reviewer #2: Yes

2. Has the statistical analysis been performed appropriately and rigorously? 

Reviewer #1: Yes

Reviewer #2: Yes

3. Have the authors made all data underlying the findings in their manuscript fully available?

Reviewer #1: No

Reviewer #2: No

4. Is the manuscript presented in an intelligible fashion and written in standard English?

Reviewer #1: Yes

Reviewer #2: Yes

5. Review Comments to the Author

Reviewer #1: The authors evaluate the impact of joint modeling of multiple longitudinal markers compared to separate models. This is an interesting follow-up of Zellner (1962)’s work in the context of linear mixed effects models. The situations where separated models are fully efficient are described as well as the situations where joint modeling is more efficient. An application to 5 longitudinal markers from a scleroderma is proposed. The paper is interesting and well-written. My comments are below.

1. It should be clarified that the first case study is based on simulated data.

2. The relationship between markers sometimes relies only on correlation between random effects (i.e., independent error), can the authors discuss how this approach is expected to perform in terms of efficiency compared to the described approach that consider correlation between random effects and residual errors as compared to separated models? This type of relationship would also likely be considered for non-Gaussian outcomes where there is no residual error.

3. Shared parameters models are also sometimes considered for the relationship between multiple longitudinal markers, this could be discussed or at least mentioned.

4. In Scleroderma case study: “we study 581 patients who have at least 4 repeated observations for each of the 5 measurements” I don’t understand why the authors make this selection? Also it is not clear if the number of observation for each marker is different and if so, how they differ. Some additional description of the data would help.

5. In cases where the separated models are fully efficient, is there any cost or benefit of fitting the combined model (other than computation time)? In practice, do the authors find exactly the same estimates for both the separated and combined models in this situation?

6. For the scleroderma study, the authors state that “measures are captured at the same or similar times”, meaning that the separated models are not fully efficient (but almost), is it likely the reason why the combined model seems to perform slightly better?

7. “RVSP and mRSS are transformed by multiplying them by -1 so that an increase in all five measures indicates improved disease status.” Is it really necessary?

8. In the subsection “Comparing bias and efficiency” of scleroderma case study section, there are two references to sections without number.

9. Discussion (p.11, line 391): patteRns

10. Appendix 0.1 (line 431): i=1,…,K should be k=1,…,K?

11. In Appendix 0.2 there is an issue with the estimation of the bias component for both the separated and combined models as the last line contains an open bracket that is not closed. Both expressions seem incomplete.

12. Appendix 0.3 and 0.4 should have different names.

Reviewer #2: This work is necessary since the joint model is considered as default in most cases and it is nice to see that the separate models work fine if fixed effects are the focus. Please include a paragraph in the discussion on what would be the case under model misspecification. Do the results only hold under well specified models?

6. PLOS authors have the option to publish the peer review history of their article (what does this mean?). If published, this will include your full peer review and any attached files.

Reviewer #1: No

Reviewer #2: **Yes: **Janet Van Niekerk

---

## [Author Response · Author response to Decision Letter 1]

8 Jan 2025

Response to Reviewer 1

Reviewer #1: The authors evaluate the impact of joint modeling of multiple longitudinal markers compared to separate models. This is an interesting follow-up of Zellner (1962)’s work in the context of linear mixed effects models. The situations where separated models are fully efficient are described as well as the situations where joint modeling is more efficient. An application to 5 longitudinal markers from a scleroderma is proposed. The paper is interesting and well-written. My comments are below.

1. It should be clarified that the first case study is based on simulated data.

Response: Thank you for pointing this out. We have clarified this point in the Case studies section (p.5, lines 154-157):

“As detailed below, the first is the general bivariate case with fixed predictors, covariance matrices, degrees of missing data, and simulated missing data patterns in which we can examine the entire space of correlations between the two measures.”

2. The relationship between markers sometimes relies only on correlation between random effects (i.e., independent error), can the authors discuss how this approach is expected to perform in terms of efficiency compared to the described approach that consider correlation between random effects and residual errors as compared to separated models? This type of relationship would also likely be considered for non-Gaussian outcomes where there is no residual error.

Response: Thank you for raising this interesting distinction about the residual error covariance matrix (Σ_Ci) being diagonal or not. A diagonal Σ_Ci is a special case of the model specification in which we assume random effects are the only source of across-measure correlations. In the bivariate case study, we calculated MSE ratios of the combined to the separated models in estimating random effects as a function of ρ_b and ρ_r under various cases of missingness and degrees of heteroskedasticity in the variance components. In figure 1, the reader can see a substantial variation in relative efficiencies of the two models as only the across-measure correlation between the random effects (ρ_b), by keeping the across-measure correlation of residual errors (ρ_r) fixed. A diagonal Σ_Ci corresponds to ρ_r=0, and even when we have a small absolute correlation of ρ_r=-0.1 or ρ_r=0.1, the combined model can be 0% to 60% more efficient depending on ρ_b. When ρ_r=0, we can expect greater efficiency when the |ρ_b | is high, especially when there is a large relative missingness of the measure of interest.

We added the following sentence in Gains in efficiencies section to summarize this finding (p. 7, lines 242-244).

“Additionally, in a special case of the model specification in which we assume random effects are the only source of across-measure correlations, we can still expect greater efficiency gains when the |ρ_b | is high.”

3. Shared parameters models are also sometimes considered for the relationship between multiple longitudinal markers, this could be discussed or at least mentioned.

Response: Thank you for the suggestion. To address this, we included findings from the paper Analyzing Multiple Outcomes: Is it Really Worth the use of Multivariate Linear Regression? by Oliveira and Teixeira-Pinto that extends the special case of the SUR result (estimates from separate regressions by OLS are fully efficient when the predictors for each outcome are the same) to parameter estimates for both shared covariates and outcome specific covariates. The authors found that the SUR result holds when estimating shared regression parameters even in the presence of outcome specific covariates in the model. If the predictors of interest are the shared parameters, separated models are fully efficient. If outcome-specific predictors are of interest, combined model should be fit unless the correlation between the measures are close to zero.

We included the following in the Introduction (p. 3, lines 69-73):

“Oliveira and Teixeira-Pinto further investigated the case in which some predictors are shared across the outcomes while others are outcome-specific and showed that the estimates for the regression parameters of the shared predictors are fully efficient while those of outcome-specific predictors have greater efficiency when a joint model is fit.”

4. In Scleroderma case study: “we study 581 patients who have at least 4 repeated observations for each of the 5 measurements” I don’t understand why the authors make this selection? Also it is not clear if the number of observation for each marker is different and if so, how they differ. Some additional description of the data would help.

Response: This selection is based on clinicians’ preference to include patients who are routinely seen in the clinic, but the method is generalizable to subjects with fewer observations.

Thank you for your suggestion about adding descriptions of the data. We added the paragraph below in the scleroderma case study section (p. 8 lines 270-275) to better describe the data and Table 1 showing the distribution of measure-specific number of observations.

“Some measures are collected more frequently than other measures. On average, we see greater numbers of skin and pulmonary measures compared to those of cardiac measures. This is because pFVC and pDLCO from pulmonary function tests and mRSS from routine clinical visits are more easily collected compared to EF and RVSP from echocardiograms. The summary statistics of the number of observations by measure are shown in Table 1.”

5. In cases where the separated models are fully efficient, is there any cost or benefit of fitting the combined model (other than computation time)? In practice, do the authors find exactly the same estimates for both the separated and combined models in this situation?

Response: Thank you for raising this point. We expect similar population level estimates if the separated models are nearly fully efficient (it would never be completely efficient unless there is zero across-measure correlation for both random effects and random errors). We do observe such patterns in other datasets when it comes to estimating fixed effects, in which case fitting separated models is preferrable. In estimating the random effects, the combined model still performs better if the amount of data differs greatly across different measures even when across-correlation is low. This is a common setting for longitudinal cohorts when some measures are collected more frequently than others.

Possible costs of fitting the combined model include having to estimate covariance terms that are near zero that is likely to result in slower convergence and greater difficulty obtaining reliable parameter estimates. However, the combined model can answer questions about across-measure correlation, which cannot be answered by fitting the separated models, even if the correlations are close to zero.

We added a summary of this to the introduction. (p. 2, lines 25-28)

6. For the scleroderma study, the authors state that “measures are captured at the same or similar times”, meaning that the separated models are not fully efficient (but almost), is it likely the reason why the combined model seems to perform slightly better?

Response: If there were no random effects in the models, the fixed effect estimates from the combined model will not be any more efficient when the measurements are captured at the same times based on the SUR results. When the measurements are captured at similar times, the combined model will perform just slightly better. Since our model includes random effects, we still observe the combined model being more efficient than the separated model.

7. “RVSP and mRSS are transformed by multiplying them by -1 so that an increase in all five measures indicates improved disease status.” Is it really necessary?

Response: Thank you for pointing this out. The reason we multiplied the two measures by –1 was that there was a clinical need to generate a composite measure as a function of the five measures based on the combined model. Higher values of this composite measure would indicate a better state of health, lower values worse. This was not a necessary procedure for the analysis included in this manuscript and removing this preprocessing step would not change the result, but we still included the step for better clinical interpretability of the pairwise correlation plot (Figure 3).

8. In the subsection “Comparing bias and efficiency” of scleroderma case study section, there are two references to sections without number.

Response: Thank you for finding the errors. We fixed them.

9. Discussion (p.11, line 391): patteRns

Response: Thank you for catching the typo. We corrected it.

10. Appendix 0.1 (line 431): i=1,…,K should be k=1,…,K?

Response: We sincerely appreciate your careful review of the manuscript. We corrected the typo in the Appendix, which is now Supporting Materials. All modifications are highlighted in the track changes version of the Supporting Materials.

11. In Appendix 0.2 there is an issue with the estimation of the bias component for both the separated and combined models as the last line contains an open bracket that is not closed. Both expressions seem incomplete.

Response: Thank you for catching the typo from a thorough review. We added the missing close brackets.

12. Appendix 0.3 and 0.4 should have different names.

Response: Thank you for pointing out the mistake. We changed the title of Appendix 0.4 (now Supporting Materials S4) to “Efficiency gains for the random effects in the case of drop-out missing pattern.”

Response to Reviewer 2

Reviewer #2: This work is necessary since the joint model is considered as default in most cases and it is nice to see that the separate models work fine if fixed effects are the focus. Please include a paragraph in the discussion on what would be the case under model misspecification. Do the results only hold under well specified models?

Response: Thank you for the suggestion.

The results in this paper describe the efficiency costs of misspecifying the covariance structure among the random effects and/or residual errors. Another form of misspecification is by omitting key predictors. In our particular application, this might involve assuming a smooth trajectory for a biomarker when the changes are more acute or immediate. The efficiency results presented in this paper assumed that linear predictors are correctly specified. When the models are misspecified, there is no a priori reason to believe that the effects of misspecification would be more or less for the combined versus separated models.

We included the above paragraph as a limitation in the discussion section (p. 12, lines 437-443).

---

## [Decision Letter · Decision Letter 1]

19 Feb 2025

Separated or joint models of repeated multivariate data to estimate individuals' disease trajectories with application to scleroderma

PONE-D-24-36884R1

Dear Dr. Kim,

We’re pleased to inform you that your manuscript has been judged scientifically suitable for publication and will be formally accepted for publication once it meets all outstanding technical requirements.

Kind regards,

Pinaki Sarder

Academic Editor

PLOS ONE

Additional Editor Comments (optional):

Reviewers' comments:

Reviewer's Responses to Questions

**Comments to the Author**

1. If the authors have adequately addressed your comments raised in a previous round of review and you feel that this manuscript is now acceptable for publication, you may indicate that here to bypass the “Comments to the Author” section, enter your conflict of interest statement in the “Confidential to Editor” section, and submit your "Accept" recommendation.

Reviewer #1: All comments have been addressed

Reviewer #2: All comments have been addressed

2. Is the manuscript technically sound, and do the data support the conclusions?

Reviewer #1: Yes

Reviewer #2: Yes

3. Has the statistical analysis been performed appropriately and rigorously? 

Reviewer #1: Yes

Reviewer #2: Yes

4. Have the authors made all data underlying the findings in their manuscript fully available?

Reviewer #1: No

Reviewer #2: Yes

5. Is the manuscript presented in an intelligible fashion and written in standard English?

Reviewer #1: Yes

Reviewer #2: Yes

6. Review Comments to the Author

Reviewer #1: (No Response)

Reviewer #2: I am satisfied with the changes. The authors have added the necessary limitation and insight into my question.

7. PLOS authors have the option to publish the peer review history of their article (what does this mean?). If published, this will include your full peer review and any attached files.

Reviewer #1: No

Reviewer #2: **Yes: **Janet Van Niekerk

---

## [Editor Report · Acceptance letter]

PONE-D-24-36884R1

PLOS ONE

Dear Dr. Kim,

I'm pleased to inform you that your manuscript has been deemed suitable for publication in PLOS ONE. Congratulations! Your manuscript is now being handed over to our production team.

Kind regards,

on behalf of

Dr. Pinaki Sarder

Academic Editor

PLOS ONE